# Neuraminidase-1: A Sialidase Involved in the Development of Cancers and Metabolic Diseases

**DOI:** 10.3390/cancers14194868

**Published:** 2022-10-05

**Authors:** Kévin Toussaint, Aline Appert-Collin, Hamid Morjani, Camille Albrecht, Hervé Sartelet, Béatrice Romier-Crouzet, Pascal Maurice, Laurent Duca, Sébastien Blaise, Amar Bennasroune

**Affiliations:** 1UMR 7369, Matrice Extracellulaire et Dynamique Cellulaire (MEDyC), Université de Reims Champagne Ardenne (URCA), 51687 Reims, France; 2Unité BioSpecT, EA7506, Université de Reims Champagne Ardenne (URCA), 51096 Reims, France

**Keywords:** Neuraminidase-1, cancer, metabolic disorders, sialidase activity

## Abstract

**Simple Summary:**

Cancers and metabolic diseases represent a leading cause of death in both developing and developed countries. Thus, new diagnostic and prognostic targets are urgently required. NEU-1 is a sialidase which regulates many membrane receptors through desialylation which results in either the activation or inhibition of the receptors. At the plasma membrane, NEU-1 is the catalytic subunit of the elastin receptor complex. This sialidase is not only required for the biological effects that are mediated by the elastin-derived peptides on several metabolic disorders, but NEU-1 is also involved in the development of various cancers. The aim of this review is to describe the role of NEU-1 in several metabolic diseases and cancers and to show that this protein could be considered in some cases as a link between these two physiopathological contexts. Consequently, NEU-1 could represent a common pharmacological target to treat putative metabolic syndrome-associated cancers like colorectal, hepatocellular and postmenopausal breast cancer.

**Abstract:**

Sialidases or neuraminidases (NEU) are glycosidases which cleave terminal sialic acid residues from glycoproteins, glycolipids and oligosaccharides. Four types of mammalian sialidases, which are encoded by different genes, have been described with distinct substrate specificity and subcellular localization: NEU-1, NEU-2, NEU-3 and NEU-4. Among them, NEU-1 regulates many membrane receptors through desialylation which results in either the activation or inhibition of these receptors. At the plasma membrane, NEU-1 also associates with the elastin-binding protein and the carboxypeptidase protective protein/cathepsin A to form the elastin receptor complex. The activation of NEU-1 is required for elastogenesis and signal transduction through this receptor, and this is responsible for the biological effects that are mediated by the elastin-derived peptides (EDP) on obesity, insulin resistance and non-alcoholic fatty liver diseases. Furthermore, NEU-1 expression is upregulated in hepatocellular cancer at the mRNA and protein levels in patients, and this sialidase regulates the hepatocellular cancer cells’ proliferation and migration. The implication of NEU-1 in other cancer types has also been shown notably in the development of pancreatic carcinoma and breast cancer. Altogether, these data indicate that NEU-1 plays a key role not only in metabolic disorders, but also in the development of several cancers which make NEU-1 a pharmacological target of high potential in these physiopathological contexts.

## 1. Introduction

The tissue-specific cell behavior is influenced by biochemical and biophysical properties of the extracellular matrix (ECM). The molecules that comprise the ECM of each tissue, including collagens, elastin, proteoglycans, laminins and fibronectin, and the way in which they are assembled establish the structure and the organization of the resultant ECM. Elastin is responsible for tissues elasticity, and this constitutes the main component of elastic fibers [1]. During aging, the elastic fibers are submitted to a degradation-induced process by a mechanical stress and by enzymatic action (matrix metalloproteinases, serine proteases or cysteine protein proteases), thereby resulting in elastin-derived peptide (EDPs) production [2,3,4,5]. The bioactive EDPs, which are also named elastokines, induce several biological effects thanks to their binding to a singular cell surface receptor called an elastin receptor complex (ERC) [6]. The ERC is a heterotrimer that is composed of a peripheral subunit named the elastin-binding protein (EBP), which binds the elastin peptides, a protective protein or cathepsin A (PPCA), and the transmembrane neuraminidase-1 (NEU-1) [7].

Neuraminidases, which are also called sialidases, are glycosidases which cleave terminal sialic acid residues from glycoproteins, glycolipids and oligosaccharides [8]. These exoglycosidases are broadly distributed among species and are notably found in viruses, protozoa, bacteria, fungi and vertebrates [9]. Four types of mammalians sialidases, which are encoded by different genes, have been described: NEU-1, NEU-2, NEU-3 and NEU-4 [10]. Each member of this family is characterized by its distinct substrate specificity and its subcellular localization. Sialidases have been involved in a broad range of human disorders, including neurodegenerative disorders, cancers and infectious and cardiovascular diseases [11].

Among these sialidases, NEU-1 plays a key role in these adverse biological effects. It is now well established that NEU-1 is not only localized in the lysosomes [12], but also at the plasma membrane where this enzyme regulates through the desialylation of the activation level of various receptors such as receptor tyrosine kinases (RTKs), integrins or Toll-like receptors [13,14,15,16,17,18,19,20,21]. The role of NEU-1 is notably associated with signal transduction through the ERC, but also to the biological effects that are induced by the EDPs on several pathologies as cancers, atherosclerosis, thrombosis, insulin resistance and non-alcoholic steatohepatitis [15,21,22,23,24,25]. Altogether, these studies indicate that NEU-1 plays a key role not only in the development of several cancers but also in metabolic disorders, which make NEU-1 a pharmacological target of high potential in these physiopathological contexts (Figure 1) [26,27,28,29]. In this review, we will focus on the role of NEU1 in several cancers and in metabolic diseases, which leads this protein to be a very interesting target in these physiopathological contexts.

## 2. Role of NEU-1 in Cancers

Cancer is a leading cause of death in both developing and developed countries. In 2020, the global incidence (newly diagnosed cases) of cancer was estimated to about 18.6 million, and almost 10.0 million cancer deaths occurred worldwide [30]. Thus, new diagnostic and prognostic targets are urgently required. In the last decade, numerous studies have been carried out to identify novel biomarkers. Many works have shown that the ECM components can play a key role in the development and progression of cancer. Elastin is an essential component of the ECM, and it is constantly modified during tumor development. As described above, during aging, elastic fibers are degraded and they generate elastin degradation products, which are also known as EDPs.

The EDPs that accumulate throughout aging promote tumor development by regulating cell proliferation, survival, invasion and angiogenesis as well as MMP expression [31]. During a tumor’s progression, the ECM is degraded by proteases generating bioactive EDPs which influence various processes that are involved in the cancer’s progression. Several studies have also described that EDPs participate in the maintenance of metastasis by supporting favorable conditions for metastatic niche development [31]. Thus, since the EDP effects are linked to the ERC complex and that the EDPs play a role in the invasive and metastatic capacities of tumors, we will describe, in this part, the role of NEU-1 in the onset and the progression of cancers (Table 1). 

### 2.1. NEU-1 and Hepatocellular Cancer

In 2018, data from GLOBOCAN showed that liver cancer is one of the most diagnosed cancers that is associated with high mortality [42]. About 70% to 90% of liver cancer cases are hepatocellular carcinomas (HCC) [43]. Most of the HCC patients are diagnosed at intermediate or advanced stages of it which prevents their surgical treatment. Indeed, only 30% of the diagnosed HCC patients receive a surgery treatment [43]. Several drugs that target the proteins that are involved in cell cycle or growth factors are currently used to treat HCC [44]. However, because of the drugs’ resistance and their side effects, the global therapeutic efficacy of these molecules is not sufficient [45]. That is why new biomarkers that are able to detect earlier and to better diagnose HCC remain necessary.

For this purpose, Hou and collaborators demonstrated the role of NEU-1 as an oncogene. Indeed, they showed that NEU-1 is upregulated in HCC at the mRNA and protein levels in two different cohort of patients. The NEU-1 expression correlates with important clinicopathological features of HCC and it is associated with the survival time. In vitro, functional studies have demonstrated that the downregulation of NEU-1 expression decreased the rate of HCC cell proliferation and migration [32].

In addition, NEU-1 has been shown to be associated with HCC that is induced by a chronic hepatitis B virus (HBV) infection [34]. Indeed, hepatitis B’s core protein which is expressed in HBV-infected hepatoma cells increased the NEU-1 expression that contributes to the epithelial–mesenchymal transition. NEU-1, therefore, promotes hepatoma cell proliferation and migration which is mediated by the HBc protein, which in return upregulates NEU-1 which is contributing to the HCC development. These data suggest that NEU-1 plays a significant role in HBV-induced HCC [35].

Recently, Wu and coworkers [33] performed a bioinformatical analysis with the ONCOMINE, GEO and TCGA datasets to assess the role of NEU-1 in HCC. Their results showed that NEU-1 is highly expressed in HCC tissues in comparison to that which is seen in normal tissues, as it has already been shown by previous studies [46,47]. Moreover, a high NEU-1 expression is correlated with advanced cancer stages or grades and a notably shorter overall survival for all HCC patients. In this study, they also demonstrated, by performing bioinformatical analyses, an implication of NEU-1 in various biological processes and molecular functions, and the correlation of some of these functions with tumorigenesis (e.g., autophagy, non-coding RNA transcripts (ncRNA) metabolic process and the regulation of cell cycle process) [33].

Altogether, these data indicate that NEU-1 plays a key role in the development and the amplification of HCC, and it can constitute a potential biomarker for HCC patients.

### 2.2. NEU-1 and Pancreatic Cancer

Pancreatic cancer is the seventh leading cause of cancer death worldwide, with a low 5-year survival rate (less than 10%). Surgery is the primary treatment for resectable pancreatic cancer, but the tumor is often diagnosed at an advanced stage of pancreas cancer and surgery can be performed in only very few cases. The identification of the risk factors and early detection markers is a crucial step that must be made to improve patient care [48].

Haxho and coworkers have reported that NEU-1 plays a key role in the sialidase-mediated tumorigenesis in pancreatic cancer [49]. Indeed, they demonstrated that the activation of the Epidermal Growth Factor Receptor (EGFR) by its ligand (EGF) induced the formation of a complex that consists of endogenous NEU-1 and matrix metalloproteinase-9 (MMP-9) which are fixed at the ectodomain of the EGFRs on the cell surface. The EGF binding to its membrane receptor induces a conformational change of the receptor and initiates the MMP-9 activation, which is followed by NEU-1 activation. Activated NEU-1 desialylates the α-2,3-sialyl residues which are linked to β-galactosides on the EGFR, and this is a determinant for EGF-induced receptor activation. Finally, they demonstrated that an increased NEU-1 expression was essential in MMP-9-EGFR signaling and it promotes cancer progression and metastasis [36].

Oseltamivir (Tamiflu) is an FDA-approved NEU inhibitor that is used for the prevention and treatment of influenza A and B infections [50]. The authors of previous works have realized that there is a possibility of treating cancer with oseltamivir. O’Shea’s team showed that oseltamivir inhibits the NEU-1 activity and suppresses the intrinsic signaling that promotes the cell survival of human pancreatic cancer (PANC1) cells that are resistant to drug therapy [51]. Moreover, the inhibition of NEU-1 by aspirin and celecoxib, which are two non-steroidal anti-inflammatory drugs, regulates the EGF-induced growth receptor activation and induces the apoptosis and necrosis processes in a dose- and time-dependent manner, thereby highlighting an important role of NEU-1 in cancer cell survival [37].

However, another work from Bera and collaborators in a gemcitabine-resistant pancreatic ductal adenocarcinoma (PDAC) cell line model has demonstrated that the miR-125b expression is increased in both the EMT and chemoresistance phenomena in part by attenuating the NEU-1 expression, thereby suggesting that there is an antitumor activity of NEU-1 [38].

Complementary studies are still necessary to achieve a better understanding of the NEU-1 effects on inhibiting the invasion process versus its role in EGFR signaling.

### 2.3. NEU-1 and Colorectal Cancer

Colorectal cancer (CRC) is the second leading cause of cancer death in both of the sexes [30]. The high mortality rate in the CRC patients is directly correlated to its ability to develop the metastasis that is reported in 50% of patients after they have undergone surgery [52]. The identification of the influencing factors and the molecular mechanisms in CRC progression could therefore improve patient survival. 

The evaluation of human NEU-1 expression by a quantitative RT-PCR in colon cancer showed that there is a lower expression of it in cancer tissues than there is in the adjacent non-cancerous mucosa. Interestingly, the NEU-1 activity level in the same cancer tissues seemed to be inversely correlated with the extent of the invasion and a poor differentiation [39]. Moreover, the NEU-1 overexpression in colon adenocarcinoma HT-29 cells decreased the cell migration and the invasion of the human CRC cells [16]. Conversely, NEU-1 knockdown increased the cell migration and invasion. In vivo, liver metastasis was significantly reduced when NEU-1 was overexpressed. The authors found that NEU-1 is able to decrease the β4-integrin sialylation which thereby reduces the phosphorylation of FAK and ERK1/2 pathway and downregulates the matrix metalloproteinase-7 activity, and then, this suppresses the β4-integrin-dependent cell migration, invasion and adhesion [16]. In addition, the study of the NEU-1 expression level in 26 different mouse colon adenocarcinoma cells which had different metastatic potential showed that the NEU-1 mRNA was less expressed, and its activity levels were lower in the highly metastatic NL17 and NL22 cells when they were compared to the NL4 and NL44 cell lines with a low metastatic potential [39].

Taken together, these data suggest that in CRC, NEU-1 overexpression is negatively associated with CRC cell invasion.

### 2.4. NEU-1 and Other Cancers

The EDPs also play an active role in the development of breast cancer [53]. In invasive breast tumors, elastolysis can be correlated with the disease’s severity. Indeed, the EDPs upregulate the MMP-14 and -2 expressions, thereby resulting in an increase in the rate of cell invasion [22]. As the biological effects of EDPs are linked to the NEU-1 activity, several studies have evaluated the role of NEU-1 in breast carcinoma cells. The NEU-1 inhibition by oseltamivir phosphate or its downregulation can impair the proliferation, apoptosis and epithelial-mesenchymal transition of breast cancer cells, and then, it can change the sialic acid level [40]. 

In vivo, studies have served to demonstrate that the treatment of xenograft mouse models of triple negative breast cancer cells by oseltamivir phosphate delayed the tumor’s growth [49]. Taken together, this literature suggests that NEU-1 plays an important role in breast cancer progression.

In addition, the effects of siRNAs which are directed against NEU-1 have been studied in ovarian cancer proliferation, apoptosis and invasion. NEU-1 is highly expressed in ovarian cancer cells in comparison with adjacent healthy tissues. The inhibition of NEU-1 expression decreases the cell proliferation, migration and invasion and cancer metastasis of several ovarian cancer cells, thereby suggesting that NEU-1 could be also a key target for the treatment of ovarian cancers [41].

In summary, the reports in this review suggest that NEU-1 could be a novel therapeutic target in cancer therapy.

## 3. Role of NEU-1 in Metabolic Diseases

The prevalence of pathological obesity, type 2 diabetes (T2D) and non-alcoholic fatty liver disease (NAFLD), which contribute to the metabolic syndrome (MS), have doubled over the past 20 years. NAFLD is approximately present in 25% of the general population and in 80% of obese or in 47–64% of T2D patients and leads to cirrhosis and hepatocellular carcinoma [54]. In the United States, MS as it is described by the International Diabetes Federation (IDF) affects 39% of the population [55], while in Europe the prevalence of MS is lower but it remains highly variable. Depending on the country, its prevalence reaches 16% of the population in Denmark in comparison to a prevalence of 21% in Ireland [56]. In addition, the World Health Organization (WHO) estimates that in 2030, the number of people that are currently suffering from MS will be multiplied by three. Besides, MS has been consistently and positively associated with the risk of very common cancers like colorectal, endometrial and postmenopausal breast cancer [57]. Therefore, the explosion of the number of people that are affected by MS may aggravate cancer progression in the next years. As previously described, NEU-1 is not only involved in the development of several cancers, but several studies have shown that this enzyme can also be associated with metabolic diseases. In this part, the implication of sialidases, and notably NEU-1, in several metabolic diseases will be described.

### 3.1. NEU-1 and Obesity

The survival and progression of tumor cells strongly depends on their ability to receive, store and use energy from glucose and/or lipids catabolism. This energy comes from its close environment and from the peritumoral adipocytes. The adipocyte manages the energy fluctuations that are induced by food intake or periods of fasting by storing energy as triglycerides or releasing it as fatty acids. In addition, the adipocyte is an active endocrine cell that secretes different molecules depending on its differentiation degree. These molecules, which are called adipokines, are involved in the regulation of the appetite and energy balance, lipid metabolism, insulin sensitivity and blood pressure regulation. Many studies have shown that adipocytes increase the tumor’s aggressiveness [58,59,60]. For example, in women, the risk of cancer is significantly increased during the menopause, a period during which a weight gain of 2–2.5 kg is very commonly recorded. With the growing number of younger and younger overweight or obese patients around the world, the risk of cancer could be rapidly increasing. The identification of new biomarkers that are in favor of an increase in the number (hyperplasia) or size (hypertrophy) of white adipocytes and new therapeutic strategy to inhibit it are, therefore, crucial.

Obesity has been shown to be associated with the modulation of sialidase activity in human leukocytes [61] and in the epididymal fat and livers in mice [62]. More recent studies that have used the broad spectrum sialidase inhibitor DANA (2-deoxy-2,3-dehydro-N-acetylneuraminic acid) have shown that there is a significant decrease in the body mass of mice that are fed with a high-fat diet [63] or with a diet that is deficient in methionine choline [64]; this model of NASH resulted in the rapid induction of liver steatosis and inflammation. This decrease could be explained by there being a reduction in the adipocyte size and in the white fat inflammation. Interestingly, the use of NEU-3 deficient-mice does not seem to be sufficient to reduce the obesity that is induced by high fat diet. These data suggest that another sialidase such as NEU-1 could play a role in adipocyte hypertrophy.

As mentioned earlier, neuraminidase can be located alone at the membrane or in a complex with ERC. This complex is activated by the stimulation of elastin-derived peptides (EDP). Several clinical or in vivo studies have showed that there are high plasma EDP concentrations in obese or diabetic patients [65,66], thereby suggesting that these EDPs could play an important role in adipose tissue remodeling. Within the laboratory, we were interested in the physiological process that allows the white adipose tissue to perform its main function of lipid accumulation, namely, adipocyte differentiation. After six day of a treatment with κ-elastin, the adipocyte differentiation and the expression of the adipogenesis transcriptional factor including C/EBPα, SREBP-1c and PPARγ were decreased [67]. The use of the ERC inhibitors, such as DANA, which is capable of blocking the EDP-induced signaling by inhibiting the NEU-1 catalytic activity [15,68] and chondroitin sulfate (CS), a galactosugar which acts like an EBP antagonist [18], allowed the promotion of adipogenesis. Indeed, the activation of the ERC sialidase activity that is mediated by the EDPs reduces the lipogenesis in the adipocytes, while the ERC inhibitors could restore the adipocyte differentiation. The binding of the EDPs on the ERC modulates different signaling pathways via NEU-1 activation. The sialidase activity of NEU-1 induces the conversion of gangliosides (GM3) into lactosyl Ceramide (lacCer), which is an early molecular actor of ERC [69,70] and a second messenger that is capable of inducing the MAPK signaling pathway [71]. The activation of the ERK signaling pathway is involved in anti-adipogenic mechanisms [72,73] by phosphorylating PPARγ which contributes to the reduction of its transcriptional activity, and therefore, to the inhibition of adipocyte differentiation [74,75]. In the context of obesity, the ERC’s NEU-1 would have a protective role regarding the hyperplasia and adipocyte differentiation. However, these data consider only the precursor cells and not the differentiated mature cells which are present in the white adipose tissue. Indeed, the chronic EDPs’ dependent stimulation of the ERC induces, in a mouse model, an increase in their weight, their fat accumulation and the overexpression of PPARγ in the adipocytes [15], thereby suggesting that there is an effect of the ERC which is activated by EDPs on the hypertrophy of the adipocytes. Nevertheless, the mechanism that allows an excessive entry of lipids or an increased lipogenic metabolism remains to be demonstrated. This mechanism might be independent of the glucose uptake and the insulin receptor activity. The ERC could thus stimulate the transcription factors like PPARγ in mature adipocytes. To reduce the activity of NEU-1, CS and/or DANA can be used. Considering that DANA is a non-specific neuraminidase inhibitor, this molecule can reduce the catalytic activity of NEU-1 but also the NEU-3 activity. This is why recent studies have aimed to identify the inhibitors which are analogs of DANA with a better selectivity for NEU-1 [29].

### 3.2. NEU-1 and Insulin Resistance (IRES)

Many meta-analyses show that having diabetes represents a major risk of developing cancer. Patients with type 1 diabetes, which is also called insulin-dependent diabetes, are more likely to develop cervical or stomach cancer [76]. Moreover, patients with type 2 diabetes have an increased risk of developing cancers that affect the breast, pancreas, liver, kidneys, endometrium or colon [76]. In addition, survival is much less likely for a diabetic patient than it is for a non-diabetic individual [76,77]. The mechanisms that are involved seem to be complex and multifactorial, and these include hyperinsulinemia, hyperglycemia and inflammation [76]. Thus, excess glucose could induce DNA damage. However, insulin resistance (IRES) preceding the emergence of type 2 diabetes could be a major component of cancer induction. Insulin receptor (IR) overexpression has been demonstrated as a factor promoting the proliferation and survival of breast, lung, colonic ovarian, endometrial and thyroid tumors [76].

Insulin is the central element that has a role in the regulation of the energy balance of our body by acting particularly on target organs such as the liver, muscles and fat. In the physiological post-prandial condition, this hormone stimulates the absorption of glucose in the muscle and inhibits gluconeogenesis and glucose release by the hepatic cells. In adipocytes, insulin promotes the transport of fatty acids from the bloodstream, promotes fat storage (lipogenesis) and inhibits fat catabolism (lipolysis). Therefore, insulin is qualified by some authors as the energy storage hormone [78,79,80]. On the other hand, a low level of insulin, which is observed during fasting, allows the mobilization of energy reserves and the release of glucose into the extracellular compartment.

This balance between energy storage and its disposal can quickly be disturbed by a chronic, high-energy nutritional intake, thereby generating a first phase called prediabetes or insulin resistance which leads to T2D. Thus, overeating and the uncontrolled intake of hyperenergetic elements (glucose or lipids) leads to the chronic and abundant plasma secretion of insulin by the pancreatic beta cells. The purpose of this secretion is to rapidly reduce the blood sugar level to 1 g/L in order to protect the cells and tissues from the damage (i.e., genetic) which is generated by hyperglycemia. However, this overabundance of plasma insulin will in turn promote a progressive decrease in the IR sensitivity on the target tissues which limits the entry of the energy molecules into the muscle, liver or adipose tissue. This vicious circle will result in hyperinsulinemia which means that the patient is less and less capable of preventing hyperglycemia. In view of these data, the insulin secretion by the pancreatic beta cells and the IR on the target cells are therefore, major elements in the regulation of energy molecules.

The studies on the pancreatic islets have shown that they have an intense sialidase activity. However, the role of sialidase in the pancreatic islets is poorly understood. Minami et al. [81] showed that the inhibition of sialidase activity with DANA facilitates insulin secretion by the beta cells. Moreover, in NEU-3-deficient mice which were fed ad libitum, their plasma insulin level was significantly higher than it was the in wild-type mice. This important secretion of insulin thus facilitates glucose tolerance. These new data suggest that the inhibition of the NEU-3 activity contributes to the enhancement of insulin release and that NEU-3 negatively regulates this mechanism. Finally, in NEU-3 knockout mice, their blood glucose level was equivalent to that of the wild-type mice after 24 h fasting, thereby suggesting that the NEU-3 desialylation activity could regulate insulin release depending on the blood glucose level [81]. To this day, the regulatory mechanism of this is not understood. It is important to note that other sialidases including NEU-1 do not appear to be involved in this insulin secretion process. Indeed, Dridi et al. showed that there is no difference between the NEU-1-deficient mice and the wild-type mice in their circulating blood insulin levels after overnight fasting or 30 min after an intraperitoneal glucose injection [19]. These results suggest that NEU-1 does not affect insulin secretion.

Otherwise, mice that overexpress NEU-3 exhibit an insulin resistance, particularly at the muscle level [82]. A decrease in the IR phosphorylation and the associated signaling pathway (i.e., phosphorylation of IRS-1) was noted. However, the situation observed with the NEU-3 overexpression is foundwith other sialidases such as NEU-1, a subunit of the ERC [15]. Indeed, the increased activity of NEU-1 which was stimulated by the EDPs also lead to a decrease in the IR phosphorylation and proteins of the associated signaling pathway including Akt and FOXO1. The decreased phosphorylation of the transcription factor FOXO1 and its translocalisation in nucleus have been described to induce the expression of the enzymes that are involved in the gluconeogenesis pathway such as glucose-6 phosphatase or phosphoenolpyruvate kinase [15,83]. However, the role of NEU-1 concerning its interaction with the IR seems to be quite complex. Indeed, when the sialidase activity of NEU-1 is stimulated by the EDP, the desialylation that is induced by NEU-1 acts as an IR inhibitor. On the contrary, Fougerat et al. described that the IR desialylation that was mediated by NEU-1 promotes the active conformation of the IR dimer when the sialidase is independent of the ERC [84]. A positive amplification loop is then initiated by insulin [19]. The binding of insulin to its receptor generates the NEU-1 activation, which then interacts with the IR and potentiates the associated signaling pathway. The activation or inhibition of the IR that is controlled by NEU-1 remains to be defined for now, but this could be linked to the glycosylation chain ramifications. Indeed, the chain’s composition and its location on this receptor or on possible protein partners will influence the protein folding, dimerization, ligand-receptor affinity, recycling or even the enzymatic activity. Molecular dynamic studies have shown that the sialic acid removal from the bi/tri-antennary chains has a strong impact on the glycans’ flexibility in the space [85]. Moreover, it has been demonstrated that some of the IR areas are covered by these same glycans. Thus, depending on the number of sialic acids that are present on the glycans and their structure, some protein regions could be hidden from the ligand [86]. This mechanism could play a key role in the receptor–ligand interaction by modifying the receptor conformation. With these data that have been obtained by molecular dynamics, we can assume that sialidases, whether they are paired or not with the ERC, could have as target sialic acids that are carried by different glycosylation chains, and this would then explain the IR activation or inhibition. Thus, even if the activation or inhibition of the IR that is controlled by NEU-1 remains to be clarified, several studies have shown that NEU-1 can influence, negatively or positively, insulin signaling and therefore, the IRES development [19,84].

### 3.3. NEU-1 and Non-Alcoholic Fatty Liver Diseases (NAFLD)

Liver disease is a major health problem worldwide. Indeed, non-alcoholic fatty liver diseases (NAFLD) are the most common chronic liver diseases [54]. They are associated with obesity and IRES and include a whole range of liver diseases. NAFLD concern 25% of the general population and are observed in 55% of type 2 diabetic patients and 80% of obese patients [87,88]. Fatty liver disease alone is usually mild, but the combination of fatty liver, hepatocyte damage and inflammation, which includes non-alcoholic steatohepatitis (NASH), can cause progressive fibrosis, cirrhosis and even hepatocellular carcinoma [89]. The main problems of these diseases are their asymptomatic development and the absence of real biomarkers. Although all of the patients with NAFLD have increased cardiovascular mortality, only the patients with portal fibrosis have an increased mortality risk that is related to liver disease.

The fibrosis that is observed in NASH is mediated by lipotoxicity, oxidative stress which is induced by excess free fatty acids (FFA), inflammation and IRES [90]. Due to the increasing obesity prevalence that is associated with that of IRES, NAFLD have become the leading causes of chronic liver diseases worldwide. NAFLD constitute a continuum of chronic pathologies. Thus, NAFLD include simple hepatic steatosis, i.e., an intrahepatic fat accumulation, NASH and cirrhosis which results frequently in hepatocarcinoma. Day and James have described the evolution of NAFLD in two studies [91]. The first one is steatosis, which is a reversible phenomenon that is associated with fat accumulation. The second one, NASH, corresponds to the fibrosis that is induced by lipotoxicity and massive inflammation, particularly with leukocytes and Kupffer cells recruitment.

Interestingly, a significant increase in the expression and activity of neuraminidases was demonstrated during the NAFLD progression in obese patients or obese mice fed [92,93]. This could suggest a major role of these sialidases in hepatic steatosis induction. Indeed, the sialidase activity inhibition by DANA shows a significant decrease in the lipid accumulation in the livers of mice that were fed with HFD [63] or in the db/db mice [25]. In obesity, tissues such as the intestine generate a significant amount of the GM3-type ceramide. Ceramides have been described as a major element in the induction of hepatic steatosis [94,95,96]. Several studies [92,97] have shown that NEU-1 or NEU-3 catalyze the GM3 hydrolysis to glucosylceramides which can be converted to glucosylceramides by GLB1, while GBA2 catalyzes the ceramides production from glucosylceramides [98]. These different intermediates could act as secondary messengers [70], thereby modulating the signaling pathways via AKT or MAPK and the potentially lipogenic and/or lipolytic activities. The involvement of sialidases in steatosis development has been recently investigated. Interestingly, Pilling et al. showed that there is a persistence of fat accumulation in NEU-3-deficient mice that were fed with a HFD diet, while the DANA treatment prevented lipogenesis [63]. Therefore, these results suggest that sialidases other than NEU-3 may be involved. In a recent study that was performed by Romier et al. [25], the stimulation of the NEU-1 catalytic activity by elastin peptides was shown to trigger steatosis. This intrahepatic lipid accumulation might be due to the inhibition of the LKB1-AMPK pathway. However, the role of NEU-1 in steatosis remains controversial. Indeed, Natori Y et al. have demonstrated that there is a decrease in the NEU-1 expression in the livers of obese subjects, while its expression is upregulated in the adipose tissue [62]. In addition, Hu Y et al. showed that highly expressed miRNAs during obesity would facilitate the lipid accumulation by inhibiting NEU-1 in NAFLD [99]. The negative impact of NEU-1 on the hepatic lipid metabolism could be explained by its interaction with the IR [84].

The second study of NAFLD, which was described by Day and James, corresponds to the lipotoxicity and inflammation effect that occurs on the fibrosis’ appearance [91]. At this stage, the phenomenon is irreversible, and it is called NASH. Mouse models inducing NASH are very limited. One of the most commonly used models is the mouse model in which mice are fed with a diet that is deficient in methionine and choline (MCD). This diet, for which the involved mechanisms are not fully understood, promotes fibrosis. In this model, collagen overexpression could be associated with the exacerbated inflammation that occurs through the activation of the monocyte-derived macrophages which promotes the NAFLD to NASH progression. A DANA treatment shows that there is a decrease in the macrophage content (MAC2 positive) in the livers of these mice [63]. Of note, the study that was performed by Romier et al. [25] resulted in the same findings. Indeed, an increase in the NEU-1 sialidase activity which was triggered by the EDP is associated with an important expression of inflammatory markers in the macrophages and the upregulation of collagen I and III expression. A treatment that combines DANA and EDPs decreases both the level of inflammation and fibrosis. This anti-fibrotic effect has also been observed on other fibrosis-prone tissues such as lung tissues [100]. Taken together, these studies strongly suggest that there is a putative implication of NEU-1 in NAFLD.

## 4. Conclusions

As NEU-1 is involved in metabolic diseases and several cancers, this protein could be considered in some cases as a link between these two physiopathological contexts. Consequently, NEU-1 could represent a common target that could be used to treat putative MS-associated cancers like colorectal, hepatocellular and postmenopausal breast cancer. Thus, NEU-1 represents a very interesting pharmacological target that could be used to reduce the adverse effect that are associated with these pathologies. Several strategies have been developed or are currently in development to inhibit the NEU-1 catalytic activity which consist of using interfering peptides, synthetic or natural analogs of DANA [26,27,28,101,102,103]. However, the inhibition of this sialidase must be finely regulated because its downregulation can provoke a dysregulation in the process of the degradation of the sialoglycoproteins, therefore causing an accumulation of over-sialylated metabolites, and then, sialidiosis [104].

## Figures and Tables

**Figure 1 cancers-14-04868-f001:**
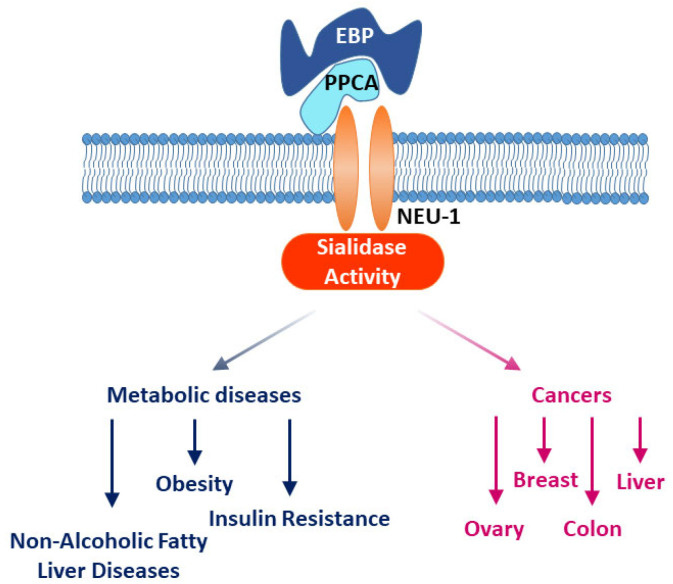
Role of NEU-1 in cancers and metabolic disorders. EBP: Elastin Binding Protein, PPCA: Protective Protein or Cathepsin A, NEU-1: Neuraminidase-1.

**Table 1 cancers-14-04868-t001:** Examples of NEU-1 implication in cancers.

Type of Cancer	Biological Effects	References
Hepatocellular Cancer	Higher mRNA and protein expression in cancer cells Correlation between NEU-1 expression and lower survival time, Cell proliferation and migration, autophagy and EMT,	[32,33,34,35]
Pancreatic Cancer	Interaction with EGFR promoting cancer progression and metastasis; Role in cancer cell survival, Chemoresistance,	[36,37,38]
Colorectal cancer	Lower expression in cancer cell; Inversely correlated with cell invasion and poor differentiation, NEU-1 overexpression negatively associated with cell invasion;	[16,39]
Breast Cancer	Role in proliferation, apoptosis and EMT,	[22,40]
Ovarian Cancer	High expression in tumor tissues; Role in cell proliferation, migration and invasion and cancer metastasis.	[41]

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
