# Peer review of "Neuraminidase-1: A Sialidase Involved in the Development of Cancers and Metabolic Diseases"

_cancers, 2022, doi:10.3390/cancers14194868_

Round 1

Reviewer 1 Report

In the review manuscript by Kévin Toussaint et. al., the authors have described the role of neuraminidases-1 (NEU1) in diseases like cancer, obesity, NAFLD, and a few others. It is appreciative and noteworthy of the authors to draft this review to put together the information on the role played by NEU1 in the progression of certain diseases. The authors have done a commendable effort on this manuscript, but there are some minor points (as follows), that authors need to pay attention to strengthen their manuscript.

1) The author describes the part played by NEU-1 in certain metabolic diseases using suitable references. The manuscript does not contain any suggestions or descriptions for how to target NEU-1 (if it is considered a potential therapeutic). The manuscript does not address the part where any small-molecule inhibitors or antibody-mediated or any other strategy can be used as a potential therapeutics. There are several published reports on the fact that NEU-1 is involved in the progression of pathological diseases and those reports also mention a possible therapeutic as a solution. In absence of any new information either on therapeutics or a more detailed biological role of NEU-1, this review manuscript seems to be a good and presentable form of reporting the published reports.

2) The authors have failed to mention the side effects of blocking, inhibiting, or targeting NEU-1. For example, inhibiting NEU-1 can cause an accumulation of sialylated glycoconjugates in the lysosomes and lead to a condition called sialidosis.

Minor typos - 

1) In Figure 1, the ligand is labeled as 'EBP'. Thus, is it 'EBP' or 'EDP' (elastin-derived peptides)?

2) At the end of line 257, a typo for a character.

Author Response

Dear reviewer,

We submitted on August 30th to Cancers the manuscript entitled " Neuraminidase-1: a sialidase involved in the development of metabolic diseases and cancers " by Toussaint et al. We fully appreciated your comments to improve the manuscript, and we modified it as requested. The majority of the issues raised by the reviewers have therefore been addressed in the manuscript and we provide a point-by-point response to the reviewers (see below).

We hope that our review article can now fulfill the criteria of this journal and be suitable for publication.

Thanking you in advance and hoping to hear from you soon,

Dr. Amar Bennasroune

Dr. Sébastien Blaise

Reviewer 1 comment: The author describes the part played by NEU-1 in certain metabolic diseases using suitable references. The manuscript does not contain any suggestions or descriptions for how to target NEU-1 (if it is considered a potential therapeutic). The manuscript does not address the part where any small-molecule inhibitors or antibody-mediated or any other strategy can be used as a potential therapeutics. There are several published reports on the fact that NEU-1 is involved in the progression of pathological diseases and those reports also mention a possible therapeutic as a solution. In absence of any new information either on therapeutics or a more detailed biological role of NEU-1, this review manuscript seems to be a good and presentable form of reporting the published reports.

Response: We agree with reviewer 1 concerning the fact that the manuscript does not address “the part where any small-molecule inhibitors or antibody-mediated or any other strategy can be used as a potential therapeutics”. However, we mentioned in this review that “several strategies have been developed or are currently in development to inhibit NEU-1 catalytic activity which consist in using interfering peptides, synthetic or natural analogs of DANA [26–28,101–103].” Indeed, the reference 103 -of the revised version of the review- (Tembely et al., 2022) corresponds to an article review of our research team in which we described the current pharmacological strategies targeting ERC activation (and notably strategies inhibiting NEU-1). That’s why we didn’t add a section concerning this point in the present review article.

Reviewer 1 comment: The authors have failed to mention the side effects of blocking, inhibiting, or targeting NEU-1. For example, inhibiting NEU-1 can cause an accumulation of sialylated glycoconjugates in the lysosomes and lead to a condition called sialidosis.

Response: In fact, this article review focus on the role of NEU-1 activation in the development of metabolic diseases and cancers which suggest that this sialidase could be a very interesting target in these physiopathological contexts. However, we totally agree with reviewer 1 concerning the fact that targeting NEU-1 can be associated with adverse effects. This sialidase inhibition must be finely regulated and that’s why we added in “Conclusions” this sentence: “However, the inhibition of this sialidase must be finely regulated because its down regulation can provoke a dysregulation in the process of degradation of sialoglycoproteins, therefore causing an accumulation of over-sialylated metabolites and then sialidiosis (Pshezhetsky et al., 1997)”

Minor typos - 

  • In Figure 1, the ligand is labeled as 'EBP'. Thus, is it 'EBP' or 'EDP' (elastin-derived peptides)?

Response: In Figure 1, the structure of the Elastin Receptor Complex (ERC) is represented. As described in introduction (and in previous publications), ERC is a heterotrimer composed of a peripheral subunit named elastin-binding protein (EBP), which binds elastin peptides, a protective protein or cathepsin A (PPCA), and the transmembrane neuraminidase-1 (NEU-1). Thus, in Figure 1, EBP corresponds to a subunit of the ERC and not to the ligand (elastin-derived peptides or EDP which are not represented). However, the meaning of the abbreviations has been added in the legend of Figure 1.

  • At the end of line 257, a typo for a character.

Response: At the end of line 257, the character has been modified: “k-elastin” is now written correctly.

Reviewer 2 Report

Dear authros,

Congratulations for developing the design of the review, for selecting only one object of study, neuramidase 1, which gives a deeper understanding of the subject. The manuscript presents a coherent reading among the topics covered.

The manuscript is a review of the mechanism of action of neuramidase 1. The review reports current works, updating the reader on what has been done about the action of neuroamidase 1. The manuscript also correlates cancer neuramidase 1 and other pathophysiologies that have inflammatory processes in common.

The manuscript first describes neuramidase in cancer and then in metabolic diseases. I suggest authors change the title to, Neuraminidase-1: a sialidase involved in the development of  cancers and metabolic diseases.

Improve the image quality of figure 1.

Line 281-28. “Considering that DANA is a non-specific neuraminidase inhibitor, 281 this molecule can reduce the catalytic activity of NEU-1 but also NEU-3 activity”.

I suggest rewriting the sentence or adding after the sentence the inhibitor specificity for NEU-1 from the group of professor Christopher Cairo has a library with modifications in the DANA structure (see doi:10.1021/acs.jmedchem.8b01411).

Author Response

Dear reviewer,

We submitted on August 30th to Cancers the manuscript entitled " Neuraminidase-1: a sialidase involved in the development of metabolic diseases and cancers " by Toussaint et al. We fully appreciated your comments to improve the manuscript, and we modified it as requested. The majority of the issues raised by the reviewers have therefore been addressed in the manuscript and we provide a point-by-point response to the reviewers (see below).

We hope that our review article can now fulfill the criteria of this journal and be suitable for publication.

Thanking you in advance and hoping to hear from you soon,

Dr. Amar Bennasroune

Dr. Sébastien Blaise

Reviewer 2 comment: “The manuscript first describes neuramidase in cancer and then in metabolic diseases. I suggest authors change the title to, Neuraminidase-1: a sialidase involved in the development of cancers and metabolic diseases.”

Response: First of all, we would like to thank reviewer 2 for his very interesting comments. As suggested by reviewer 2, we replaced the title “Neuraminidase-1: a sialidase involved in the development of metabolic diseases and cancers” by “Neuraminidase-1: a sialidase involved in the development of cancers and metabolic diseases”.

Reviewer 2 comment: “Improve the image quality of figure 1.”

Response: The image quality of Figure 1 has been improved as well as possible.

Reviewer 2 comment: “Line 281-28. “Considering that DANA is a non-specific neuraminidase inhibitor, this molecule can reduce the catalytic activity of NEU-1 but also NEU-3 activity”. I suggest rewriting the sentence or adding after the sentence the inhibitor specificity for NEU-1 from the group of professor Christopher Cairo has a library with modifications in the DANA structure (see doi:10.1021/acs.jmedchem.8b01411).

Response: As suggested by reviewer 2, we added this sentence: “This is why recent studies were aimed to identify inhibitors which are analogs of DANA with a better selectivity for NEU-1 [29].”
